# Approximate inference in latent Gaussian-Markov models from continuous time observations

**Botond Cseke**[1]               **Manfred Opper**[2]               **Guido Sanguinetti**[1]

[1]School of Informatics
University of Edinburgh, U.K.
{bcseke,gsanguin}@inf.ed.ac.uk

[2]Computer Science
TU Berlin, Germany
manfred.opper@tu-berlin.de

## Abstract

We propose an approximate inference algorithm for continuous time Gaussian Markov process models with both discrete and continuous time likelihoods. We show that the continuous time limit of the expectation propagation algorithm exists and results in a hybrid fixed point iteration consisting of (1) expectation propagation updates for discrete time terms and (2) variational updates for the continuous time term. We introduce post-inference corrections methods that improve on the marginals of the approximation. This approach extends the classical Kalman-Bucy smoothing procedure to non-Gaussian observations, enabling continuous-time inference in a variety of models, including spiking neuronal models (state-space models with point process observations) and box likelihood models. Experimental results on real and simulated data demonstrate high distributional accuracy and significant computational savings compared to discrete-time approaches in a neural application.

## 1 Introduction

Continuous time stochastic processes provide a flexible and popular framework for data modelling in a broad spectrum of scientific and engineering disciplines. Their intrinsically non-parametric, infinite-dimensional nature also makes them a challenging field for the development of efficient inference algorithms. Recent years have seen several such algorithms being proposed for a variety of models [Opper and Sanguinetti, 2008, Opper et al., 2010, Rao and Teh, 2012]. Most inference work has focused on the scenario when observations are available at a finite set of time points, however, modern technologies are making effectively *continuous time observations* increasingly common: for example, high speed imaging technologies now enable the acquisition of biological data at around 100Hz for extended periods of time. Other scenarios give intrinsically continuous time observations: for example, sensors monitoring the transit of a particle through a barrier provide continuous time data on the particle's position. To the best of our knowledge, this problem has not been addressed in the statistical machine learning community.

In this paper, we propose an expectation-propagation (EP)-type algorithm [Opper and Winther, 2000, Minka, 2001] for latent diffusion processes observed in either discrete or continuous time. We derive fixed-point update equations by considering a continuous time limit of the parallel EP algorithm [e.g. Opper and Winther, 2005, Cseke and Heskes, 2011b]: these fixed point updates naturally become differential equations in the continuous time limit. Remarkably, we show that, in the presence of continuous time observations, the update equations for the EP algorithm reduce to updates for a variational Gaussian approximation [Archambeau et al., 2007]. We also generalise to the continuous-time limit the EP correction scheme of [Cseke and Heskes, 2011b], which enable us to capture some of the non-Gaussian behaviour of the time marginals.

## 2 Models and methods

We consider dynamical systems described by multivariate stochastic differential equations (SDEs) of Ornstein-Uhlenbeck (OU) type over the $[0, 1]$ time interval

$$d\boldsymbol{x}_t = (\boldsymbol{A}_t \boldsymbol{x}_t + \boldsymbol{c}_t)dt + \boldsymbol{B}_t^{1/2} d\boldsymbol{W}_t, \tag{1}$$

where $\{\boldsymbol{W}_t\}_t$ is the standard Wiener process [Gardiner, 2002] and $\boldsymbol{A}_t$, $\boldsymbol{B}_t$ and $\boldsymbol{c}_t$ are time dependent matrix and vector valued functions respectively with $\boldsymbol{B}_t$ being positive definite for all $t \in [0, 1]$. Even though the process does not posses a formulation through density functions (with respect to the Lebesgue measure), in order to be able to symbolically represent and manipulate the variables of the process in the Bayesian formalism, we will use the proxy $p_0(\{\boldsymbol{x}_t\})$ to denote their distribution.

The process can be observed (noisily) both at discrete time points, and for continuous time intervals; we will partition the observations in $\boldsymbol{y}_{t_i}^d$, $t_i \in T_d$ and $\boldsymbol{y}_t^c$, $t \in [0, 1]$ accordingly. We assume that the likelihood function admits the general formulation

$$p(\{\boldsymbol{y}_{t_i}^d\}, \{\boldsymbol{y}_t^c\} | \{\boldsymbol{x}_t\}) \propto \prod_{t_i \in T_d} p(\boldsymbol{y}_{t_i}^d | \boldsymbol{x}_{t_i}) \times \exp\left\{ -\int_0^1 dt V(t, \boldsymbol{y}_t^c, \boldsymbol{x}_t) \right\}. \tag{2}$$

We refer to $p(\boldsymbol{y}_{t_i}^d | \boldsymbol{x}_{t_i})$ and $V(t, \boldsymbol{y}_t^c, \boldsymbol{x}_t)$ as discrete time likelihood term and continuous time loss function, respectively. We notice that, using Girsanov's theorem and Ito's lemma, non-linear diffusion equations with constant (diagonal) diffusion matrix can be re-written in the form (1)-(2), provided the drift can be obtained as the gradient of a potential function [e.g. Øksendal, 2010].

Our aim is to propose approximate inference methods to compute the marginals $p(\boldsymbol{x}_t | \{\boldsymbol{y}_{t_i}^d\}_i, \{\boldsymbol{y}_t^c\})$ of the posterior distribution

$$p(\{\boldsymbol{x}_t\}_t | \{\boldsymbol{y}_{t_i}^d\}_i, \{\boldsymbol{y}_t^c\}) \propto p(\{\boldsymbol{y}_{t_i}^d\}_i, \{\boldsymbol{y}_t^c\} | \{\boldsymbol{x}_t\}) \times p_0(\{\boldsymbol{x}_t\}).$$

### 2.1 Exact inference in Gaussian models

We start form the exact case of Gaussian observations and quadratic loss function. The linearity of equation (1) implies that the marginal distributions of the process at every time point are Gaussian (assuming Gaussian initial conditions). The time evolution of the marginal mean $\boldsymbol{m}_t$ and covariance $\boldsymbol{V}_t$ is governed by the pair of differential equations [Gardiner, 2002]

$$\frac{d}{dt}\boldsymbol{m}_t = \boldsymbol{A}_t \boldsymbol{m}_t + \boldsymbol{c}_t \quad \text{and} \quad \frac{d}{dt}\boldsymbol{V}_t = \boldsymbol{A}_t \boldsymbol{V}_t + \boldsymbol{V}_t \boldsymbol{A}_t^T + \boldsymbol{B}_t. \tag{3}$$

In the case of Gaussian observations and a quadratic loss function $V(t, \boldsymbol{y}_t^c, \boldsymbol{x}_t) = \text{const.} - \boldsymbol{x}_t^T \boldsymbol{h}_t^c + \frac{1}{2}\boldsymbol{x}_t^T \boldsymbol{Q}_t^c \boldsymbol{x}_t$, these equations, together with their backward analogues, enable an exact recursive inference algorithm, known as the Kalman-Bucy smoother [e.g. Särkkä, 2006]. This algorithm arises because we can recast the loss function as an auxiliary (observation) process

$$d\boldsymbol{y}_t^c = \boldsymbol{x}_t dt + \boldsymbol{R}_t^{1/2} d\boldsymbol{W}_t, \tag{4}$$

where $\boldsymbol{R}_t^{-1} = \boldsymbol{Q}_t^c$ and $\boldsymbol{R}_t^{-1} d\boldsymbol{y}_t^c/dt = \boldsymbol{h}_t^c$. This follows by the Gaussianity of the observation process and the fundamental property of Ito's calculus $d\boldsymbol{W}_t^2 = \boldsymbol{I}dt$.

The Kalman-Bucy algorithm computes the posterior marginal means and covariances by solving the differential equations in a forward-backward fashion. These can be combined with classical Kalman filtering to account for discrete-time observations. The exact form of the equations as well as the variational derivation of the Kalman-Bucy problem are given in Section B of the Supplementary Material.

### 2.2 Approximate inference

In this section we use an Euler discretisation of the prior and the continuous time likelihood to turn our model into a multivariate latent Gaussian model. We review the EP algorithm for such models and then we show that when taking the limit $\Delta t \to 0$ the updates of the EP algorithm exist. The resulting approximate posterior process is again an OU process and we compute its parameters. Finally, we show how corrections to the marginals proposed [Cseke and Heskes, 2011b] can be extended to the continuous time case.

### 2.2.1 Euler discretisation

Let $T = \{t_1 = 0, t_2, \ldots, t_{K-1}, t_K = 1\}$ be a discretisation of the $[0, 1]$ interval and let the matrix $\boldsymbol{x} = [\boldsymbol{x}_{t_1}, \ldots, \boldsymbol{x}_{t_K}]$ represent the process $\{\boldsymbol{x}_t\}_t$ using the discretisation given by $T$. Without loss of generality we can assume that $T_d \subset T$. We assume the Euler-Maruyama approach and approximate $p(\{\boldsymbol{x}_t\})$ by[1]

$$p_0(\boldsymbol{x}) = N(\boldsymbol{x}_0; \boldsymbol{m}_0, \boldsymbol{V}_0) \prod_k N(\boldsymbol{x}_{t_{k+1}}; \boldsymbol{x}_{t_k} + (\boldsymbol{A}_{t_k}\boldsymbol{x}_{t_k} + \boldsymbol{c}_{t_k})\Delta t_k, \Delta t_k \boldsymbol{B}_{t_k})$$

and in a similar fashion we approximate the continuous time likelihood by

$$p(\boldsymbol{y}^c | \boldsymbol{x}) \propto \exp\left\{-\sum_k \Delta t_k V(t_k, \boldsymbol{y}_{t_k}^c, \boldsymbol{x}_{t_k})\right\},$$

where $\boldsymbol{y}^c$ is the matrix $\boldsymbol{y}^c = [\boldsymbol{y}_{t_1}^c, \ldots, \boldsymbol{y}_{t_K}^c]$. Consequently we approximate our model by the latent Gaussian model

$$p(\{\boldsymbol{y}_{t_i}^d\}_i, \boldsymbol{y}^c, \boldsymbol{x}) = p_0(\boldsymbol{x}) \times \prod_i p(\boldsymbol{y}_{t_i}^d | \boldsymbol{x}_{t_i}) \prod_k \exp\left\{-\Delta t_k V(t_k, \boldsymbol{y}_{t_k}^c, \boldsymbol{x}_{t_k})\right\}$$

where we remark that the prior $p_0$ has a block-diagonal precision structure. To simplify notation, in the following we use the aliases $\phi_i^d(\boldsymbol{x}_{t_i}) = p(\boldsymbol{y}_{t_i}^d | \boldsymbol{x}_{t_i})$ and $\phi_k^c(\boldsymbol{x}_{t_k}; \Delta t_k) = \exp\left\{-\Delta t_k V(t_k, \boldsymbol{y}_{t_k}^c, \boldsymbol{x}_{t_k})\right\}$.

### 2.2.2 Inference using expectation propagation

Expectation propagation [Opper and Winther, 2000, Minka, 2001] is a well known algorithm that provides good approximations of the posterior marginals in latent Gaussian models. We use here the parallel EP approach [e.g. Cseke and Heskes, 2011b]; similar continuous time limiting arguments can be made for the original (sequential) EP approach. The algorithm approximates the posterior $p(\boldsymbol{x}|\{\boldsymbol{y}_{t_i}^d\}_i, \boldsymbol{y}^c)$ by a Gaussian

$$q_0(\boldsymbol{x}) \propto p_0(\boldsymbol{x}) \prod_i \tilde{\phi}_i^d(\boldsymbol{x}_{t_i}) \prod_k \tilde{\phi}_k^c(\boldsymbol{x}_{t_k}; \Delta t_k),$$

where $\tilde{\phi}_i^d$ and $\tilde{\phi}_k^c$ are Gaussian functions. When applied to our model the algorithm proceeds by performing the fixed point iteration

$$[\tilde{\phi}_i^d(\boldsymbol{x}_{t_i})]^{new} \propto \frac{\text{Collapse}(\phi_i^d(\boldsymbol{x}_{t_i})\tilde{\phi}_i^d(\boldsymbol{x}_{t_i})^{-1}q_0(\boldsymbol{x}_{t_i}); \mathcal{N})}{q_0(\boldsymbol{x}_{t_i})} \times \tilde{\phi}_i^d(\boldsymbol{x}_{t_i}) \quad \text{for all } t_i \in T_d, \tag{5}$$

$$[\tilde{\phi}_k^c(\boldsymbol{x}_{t_k}; \Delta t_k)]^{new} \propto \frac{\text{Collapse}(\phi_k^c(\boldsymbol{x}_{t_k}; \Delta t_k)\tilde{\phi}_k^c(\boldsymbol{x}_{t_k}; \Delta t_k)^{-1}q_0(\boldsymbol{x}_{t_k}); \mathcal{N})}{q_0(\boldsymbol{x}_{t_k})} \times \tilde{\phi}_k^c(\boldsymbol{x}_{t_k}; \Delta t_k) \quad \text{for all } t_k \in T, \tag{6}$$

where $\text{Collapse}(p(\boldsymbol{z}); \mathcal{N}) = \text{argmin}_{q \in \mathcal{N}} D[p(\boldsymbol{z})||q(\boldsymbol{z})]$ denotes the projection of the density $p(\boldsymbol{z})$ into the Gaussian family denoted by $\mathcal{N}$. In other words, $\text{Collapse}(p(\boldsymbol{z}); \mathcal{N})$ is the Gaussian density that matches the first and second moments of $p(\boldsymbol{z})$. Readers familiar with the classical formulation of EP [Minka, 2001] will recognise in equation (5) the so-called *term updates*, where $\tilde{\phi}_i^d(\boldsymbol{x}_{t_i})^{-1}q_0(\boldsymbol{x}_{t_i})$ is the *cavity distribution* and $\phi_i^d(\boldsymbol{x}_{t_i})\tilde{\phi}_i^d(\boldsymbol{x}_{t_i})^{-1}q_0(\boldsymbol{x}_{t_i})$ the *tilted distribution*. Equations (5-6) imply that at any fixed point of the iterations we have $q(\boldsymbol{x}_{t_i}) = \text{Collapse}(\phi_i^d(\boldsymbol{x}_{t_i})\tilde{\phi}_i^d(\boldsymbol{x}_{t_i})^{-1}q_0(\boldsymbol{x}_{t_i}); \mathcal{N})$ and $q(\boldsymbol{x}_{t_k}) = \text{Collapse}(\phi_k^c(\boldsymbol{x}_{t_k}; \Delta t_k)\tilde{\phi}_k^c(\boldsymbol{x}_{t_k}; \Delta t_k)^{-1}q_0(\boldsymbol{x}_{t_k}); \mathcal{N})$. The algorithm can also be derived and justified as a constrained optimisation problem of a Gibbs free energy formulation [Heskes et al., 2005]; this alternative approach can also be shown to extend to the continuous time limit (see Section A.2 of the Supplementary Material) and provides a useful tool for approximate evidence calculations.

Equation (5) does not depend on the time discretisation, and hence provides a valid update equation also working directly with the continuous time process. On the other hand, the quantities in equation (6) depend explicitly on $\Delta t_k$, and it is necessary to ensure that they remain well defined (and computable) in the continuous time limit. In order to derive the limiting behaviour of (6) we introduce the the following notation: (i) we use $\boldsymbol{f}(\boldsymbol{z}) = (\boldsymbol{z}, -\boldsymbol{z}\boldsymbol{z}^T/2)$ to denote the sufficient statistic of a multivariate Gaussian (ii), we use $\boldsymbol{\lambda}_{\boldsymbol{t_i}}^{\boldsymbol{d}} = (\boldsymbol{h}_{t_i}^d, \boldsymbol{Q}_{t_i}^d)$ as the canonical parameters corresponding to the Gaussian function $\tilde{\phi}_i^d(\boldsymbol{x}_{t_i}) \propto \exp\{\boldsymbol{\lambda}_{t_i}^d \cdot \boldsymbol{f}(\boldsymbol{x}_{t_i})\}$[2], (iii) we use $\boldsymbol{\lambda}_{t_k}^c = (\boldsymbol{h}_{t_k}^c, \boldsymbol{Q}_{t_k}^c)$ as the canonical parameters corresponding to the Gaussian function $\tilde{\phi}_k^c(\boldsymbol{x}_{t_k}) \propto \exp\{\Delta t_k \boldsymbol{\lambda}_{t_k}^c \cdot \boldsymbol{f}(\boldsymbol{x}_{t_k})\}$, and finally, (iv) we use $\text{Collapse}(p(\boldsymbol{z}); \boldsymbol{f})$ as

the canonical parameters corresponding to the density $\text{Collapse}(p(\boldsymbol{z}); \mathcal{N})$. By using this notation we can rewrite (6) as

$$[\boldsymbol{\lambda}_{t_k}^c]^{new} = \boldsymbol{\lambda}_{t_k}^c + \frac{1}{\Delta t_k} \left[ \text{Collapse}(q_c(\boldsymbol{x}_{t_k}); \boldsymbol{f}) - \text{Collapse}(q_0(\boldsymbol{x}_{t_k}); \boldsymbol{f}) \right] \tag{7}$$

with

$$q_c(\boldsymbol{x}_{t_k}) \propto \exp(-\Delta t_k [V(t_k, \boldsymbol{x}_{t_k}) + \boldsymbol{\lambda}_{t_k}^c \cdot \boldsymbol{f}(\boldsymbol{x}_{t_k})]) q_0(\boldsymbol{x}_{t_k}). \tag{8}$$

The approximating density can then be written as

$$q_0(\boldsymbol{x}) \propto p_0(\boldsymbol{x}) \times \exp\left\{ \sum_i \boldsymbol{\lambda}_{t_i}^d \cdot \boldsymbol{f}(\boldsymbol{x}_{t_i}) + \sum_k \Delta t_k \boldsymbol{\lambda}_{t_k}^c \cdot \boldsymbol{f}(\boldsymbol{x}_{t_k}) \right\}. \tag{9}$$

By direct Taylor expansion of $\text{Collapse}(q_c(\boldsymbol{x}_{t_k}); \boldsymbol{f})$ one can show that the update equation (7) remains finite when we take the limit $\Delta t_k \to 0$. A slightly more general perspective however affords greater insight into the algorithm, as shown below.

### 2.2.3 Continuous time limit of the update equations

Let $\boldsymbol{\mu}_{t_k} = \text{Collapse}(q_0(\boldsymbol{x}_{t_k}); \boldsymbol{f})$ and denote by $Z(\Delta t_k, \boldsymbol{\mu}_{t_k})$ and $Z(\boldsymbol{\mu}_{t_k})$ the normalisation constant of $q_c(\boldsymbol{x}_{t_k})$ and $q_0(\boldsymbol{x}_{t_k})$ respectively. The notation emphasises that $q_c(\boldsymbol{x}_{t_k})$ differs from $q_0(\boldsymbol{x}_{t_k})$ by a term dependent on the granularity of the discretisation $\Delta t_k$. We exploit the well known fact that the derivatives with respect to the canonical parameters of the log normalisation constant of a distribution within the exponential family give the moment parameters of the distribution. From the definition of $q_c(\boldsymbol{x}_{t_k})$ in equation (8) we then have that its first two moments can be computed as $\partial_{\boldsymbol{\mu}_{t_k}} \log Z(\Delta t_k, \boldsymbol{\mu}_{t_k})$. The Collapse operation in (7) can then be rewritten as

$$\text{Collapse}(q_c(\boldsymbol{x}_{t_k}); \boldsymbol{f}) = \Psi(\partial_{\boldsymbol{\mu}_{t_k}} \log Z(\Delta t_k, \boldsymbol{\mu}_{t_k})), \tag{10}$$

where $\Psi$ is the function transforming the moment parameters of a Gaussian into its (canonical) parameters. We now assume $\Delta t_k$ to be small and expand $Z(\Delta t_k, \boldsymbol{\mu}_{t_k})$ to first order in $\Delta t_k$. By using the property that $\lim_{\alpha \to 0^+} \langle g(z)^\alpha \rangle_{p(z)}^{1/\alpha} = \exp(\langle \log g(x) \rangle_p)$ for any distribution $p(z)$ and $g(z) > 0$, one can write

$$\lim_{\Delta t_k \to 0} \frac{1}{\Delta t_k} [\log Z(\Delta t_k, \boldsymbol{\mu}_{t_k}) - \log Z(\boldsymbol{\mu}_{t_k})] = \log \lim_{\Delta t_k \to 0} \left\langle \exp\{-\Delta t_k [V(t_k, \boldsymbol{x}_{t_k}) + \boldsymbol{\lambda}_{t_k}^c \cdot \boldsymbol{f}(\boldsymbol{x}_{t_k})]\} \right\rangle_{q_0(\boldsymbol{x}_{t_k})}^{1/\Delta t_k}$$

$$= -\left\langle [V(t_k, \boldsymbol{x}_{t_k}) + \boldsymbol{\lambda}_{t_k}^c \cdot \boldsymbol{f}(\boldsymbol{x}_{t_k})] \right\rangle_{q_0(\boldsymbol{x}_{t_k})}$$

$$= -\left\langle V(t_k, \boldsymbol{x}_{t_k}) \right\rangle_{q_0(\boldsymbol{x}_{t_k})} - \Psi^{-1}(\boldsymbol{\mu}_{t_k}) \boldsymbol{\lambda}_{t_k}^c, \tag{11}$$

where we exploited the fact that $\langle \boldsymbol{f}(\boldsymbol{x}_{t_k}) \rangle_{q_0(\boldsymbol{x}_{t_k})}$ are the moments of the $q_0(\boldsymbol{x}_{t_k})$ distribution. We can now exploit the fact that $\Delta t_k$ is small and linearise the nonlinear map $\Psi$ about the moments of $q_0(\boldsymbol{x}_{t_k})$ to obtain a first order approximation to equation (10) as

$$\text{Collapse}(q_c(\boldsymbol{x}_{t_k}); \boldsymbol{f}) \simeq \boldsymbol{\mu}_{t_k} - \Delta t_k \boldsymbol{\lambda}_{t_k}^c - \Delta t_k J_\Psi(\boldsymbol{\mu}_{t_k}) \partial_{\boldsymbol{\mu}_{t_k}} \langle V(t_k, \boldsymbol{x}_{t_k}) \rangle_{q_0(\boldsymbol{x}_{t_k})} \tag{12}$$

where $J_\Psi(\boldsymbol{\mu}_{t_k})$ denotes the Jacobian matrix of the map $\Psi$ evaluated at $\boldsymbol{\mu}_{t_k}$. The second term on the r.h.s. of equation (12) follows from the obvious identity $\partial_{\boldsymbol{\mu}_{t_k}} \Psi(\Psi^{-1}(\boldsymbol{\mu}_{t_k})) = I$.

By substituting (12) into (7), we take the limit $\Delta t_k \to 0$ and obtain the update equations

$$[\boldsymbol{\lambda}_t^c]^{new} = -J_\Psi(\boldsymbol{\mu}_t) \partial_{\boldsymbol{\mu}_t} \langle V(t, \boldsymbol{x}_t) \rangle_{q_0(\boldsymbol{x}_t)} \quad \text{for all } t \in [0, 1]. \tag{13}$$

Notice that the updating of $\boldsymbol{\lambda}_t^c$ is somewhat hidden in equation (13); the "old" parameters are in fact contained in the parameters $\boldsymbol{\mu}_{t_k}$. Since $\boldsymbol{\lambda}_t^c$ corresponds to the canonical parameters of a multivariate Gaussian, we can use the representation $\boldsymbol{\lambda}_t^c = (\boldsymbol{h}_t^c, \boldsymbol{Q}_t^c)$ and after some algebra on the moment-canonical transformation of Gaussians we write the fixed point iteration as

$$[\boldsymbol{h}_t^c]^{new} = -\partial_{\boldsymbol{m}_t} \langle V(t, \boldsymbol{x}_t) \rangle_{q_0(\boldsymbol{x}_t)} + 2\partial_{\boldsymbol{V}_t} \langle V(t, \boldsymbol{x}_t) \rangle_{q_0(\boldsymbol{x}_t)} \boldsymbol{m}_t \quad \text{and} \quad [\boldsymbol{Q}_t^c]^{new} = \partial_{\boldsymbol{V}_t} \langle V(t, \boldsymbol{x}_t) \rangle_{q_0(\boldsymbol{x}_t)}, \tag{14}$$

where $\boldsymbol{m}_t$ and $\boldsymbol{V}_t$ are the marginal means and covariances of $q_0$ at the $\Delta t_k \to 0$. Algorithmically, computing the marginal moments and covariances of the discretised Gaussian $q_0(\boldsymbol{x})$ in (9) can be done by solving a sparse linear system and doing partial matrix inversion using the Cholesky factorisation and the Takahashi equations as in Cseke and Heskes [2011b]. This corresponds to a junction tree algorithm on a (block) chain graph [Davis, 2006] which, in the continuous time limit, can be reduced to a set of differential equations

due to the chain structure of the graph. Alternatively, one can notice that, in the continuous time limit, the structure of $q_0(\boldsymbol{x})$ in equation (9) defines a posterior process for an OU process $p_0(\{\boldsymbol{x}_t\})$ observed at discrete times with Gaussian noise (corresponding to the terms $\tilde{\phi}_i^d(\boldsymbol{x}_{t_i})$ with canonical parameters $\boldsymbol{\lambda}_{t_i}^d$) and with a quadratic continuous time loss, which is computed using equation (14). The moments therefore be computed using the Kalman-Bucy algorithm; details of the algorithm are given in Section B.1 of the Supplementary Material. The derivation above illustrates another interesting characteristic of working with continuous-time likelihoods. Readers familiar with the fractional free energies and the power EP algorithm may notice that the time lag $\Delta t_k$ plays a similar role as the fractional or power parameter $\alpha$. It is well known property that in the $\alpha \to 0$ limit the algorithm and the free energy collapses to variational [e.g. Wiegerinck and Heskes, 2003, Cseke and Heskes, 2011a] and thus, intuitively, the collapse and the existence of the limit is related to this property.

Overall, we arrive to a hybrid algorithm in which: (i) the canonical parameters $(\boldsymbol{h}_{t_i}^d, \boldsymbol{Q}_{t_i}^d)$ corresponding to the discrete time terms are updated by the usual EP updates in (5), (ii) the canonical parameters $(\boldsymbol{h}_t^c, \boldsymbol{Q}_t^c)$ corresponding to the continuous loss function $V(t, \boldsymbol{x}_t)$ are updated by the variational updates in (14) (iii), the marginal moment parameters of $q_0(\boldsymbol{x}_t)$ are computed by the forward-backward differential equations referred to in Section 2.1. We can use either parallel or a forward-backward type scheduling. A more detailed description of the inference algorithm is given in Section C of the Supplementary Material. The algorithm performs well in the comfort zone of EP, that is, log-concave discrete likelihood terms and convex loss. Non-convergence can occur in case of multimodal likelihoods and loss functions and alternative options to optimise the free energy have to be explored [e.g. Heskes et al., 2005, Archambeau et al., 2007].

### 2.2.4 Parameters of the approximating OU process

The fixed point iteration scheme computes only the marginal means and covariances of $q_0(\{x_t\})$ and it does not provide a parametric OU process as an approximation. However, this can be computed by finding the parameters of an OU process that matches $q_0$ in the moment matching Kullback-Leibler divergence. That is, if $q^*(\{\boldsymbol{x}_t\})$ minimises $D[q_0(\{\boldsymbol{x}_t\})||q^*(\{\boldsymbol{x}_t\})]$, then the parameters of $q^*$ are given by

$$\boldsymbol{A}_t^* = \boldsymbol{A}_t - \boldsymbol{B}_t[\boldsymbol{V}_t^{bw}]^{-1}, \quad \boldsymbol{c}_t^* = \boldsymbol{c}_t + \boldsymbol{B}_t[\boldsymbol{V}_t^{bw}]^{-1}\boldsymbol{m}_t^{bw} \quad \text{and} \quad \boldsymbol{B}_t^* = \boldsymbol{B}_t, \tag{15}$$

where $\boldsymbol{m}_t^{bw}$ and $\boldsymbol{V}_t^{bw}$ are computed by the backward Kalman-Bucy filtering equations. The computations are somewhat lengthy; a full derivation can be found in Section B.3 of the Supplementary Material.

### 2.2.5 Corrections to the marginals

In this section we extend the factorised correction method for multivariate latent Gaussian models introduced in Cseke and Heskes [2011b] to continuous time observations. Other correction schemes [e.g. Opper et al., 2009] can in principle also be applied. We start again from the discretised representation and then take the $\Delta t_k \to 0$. To begin with, we focus on the corrections from the continuous time observation process. By removing the Gaussian terms (with canonical parameters $\boldsymbol{\lambda}_{t_k}^c$) from the approximate posterior and replacing them with the exact likelihood, we can rewrite the exact discretised posterior as

$$p(\boldsymbol{x}) \propto q_0(\boldsymbol{x}) \times \exp\left\{ -\sum_k \Delta t_k[V(t_k, \boldsymbol{x}_{t_k}) + \boldsymbol{\lambda}_{t_k}^c \cdot \boldsymbol{f}(\boldsymbol{x}_t)]\right\}.$$

The exact posterior marginal at time $t_j$ is thus given by

$$p(\boldsymbol{x}_{t_j}) \propto q_0(\boldsymbol{x}_{t_j}) \times \exp\left\{ -\Delta t_j[V(t_j, \boldsymbol{x}_{t_j} + \boldsymbol{\lambda}_{t_j}^c \cdot \boldsymbol{f}(\boldsymbol{x}_{t_j}))]\right\} \times c_T(\boldsymbol{x}_{t_j})$$

with

$$c_T(\boldsymbol{x}_{t_j}) = \int d\boldsymbol{x}_{\backslash t_j} q_0(\boldsymbol{x}_{\backslash t_j}|\boldsymbol{x}_{t_j}) \times \exp\left\{ -\sum_{k \neq j} \Delta t_k[V(t_k, \boldsymbol{x}_{t_k}) + \boldsymbol{\lambda}_{t_k}^c \cdot \boldsymbol{f}.(\boldsymbol{x}_{t_k})]\right\},$$

where the subscript $\backslash j$ indicates the whole vector with the $j$-th entry removed. By approximating the joint conditional $q_0(\boldsymbol{x}_{\backslash t_j}|\boldsymbol{x}_{t_j})$ with a product of its marginals and taking the $\Delta t_k \to 0$ limit, we obtain

$$c(\boldsymbol{x}_t) \simeq \exp\left\{ -\int_0^1 ds \, \langle V(s, \boldsymbol{x}_s) + \boldsymbol{\lambda}_s^c \cdot \boldsymbol{f}(\boldsymbol{x}_s)\rangle_{q_0(\boldsymbol{x}_s|\boldsymbol{x}_t)}\right\}.$$

When combining the continuous part and the factorised discrete time corrections—by adding the discrete time terms to the formalism above—we arrive to the corrected approximate marginal

$$\tilde{p}(\boldsymbol{x}_t) \propto q_0(\boldsymbol{x}_t) \exp\left\{ -\int_0^1 ds \, \langle V(s, \boldsymbol{x}_s) + \boldsymbol{\lambda}_s^c \cdot \boldsymbol{f}(\boldsymbol{x}_s)\rangle_{q_0(\boldsymbol{x}_s|\boldsymbol{x}_t)}\right\} \times \prod_i \left\langle \frac{p(\boldsymbol{y}_{t_i}^d|\boldsymbol{x}_{t_i})}{\exp\{\boldsymbol{\lambda}_{t_i}^d \cdot \boldsymbol{f}(\boldsymbol{x}_{t_i})\}}\right\rangle_{q_0(\boldsymbol{x}_{t_i}|\boldsymbol{x}_t)}.$$

For any fixed $t$ one can compute the correlations in linear time by using the parametric form of the approximation in 15. The evaluations for a fixed $\boldsymbol{x}_t$ are also linear in time.

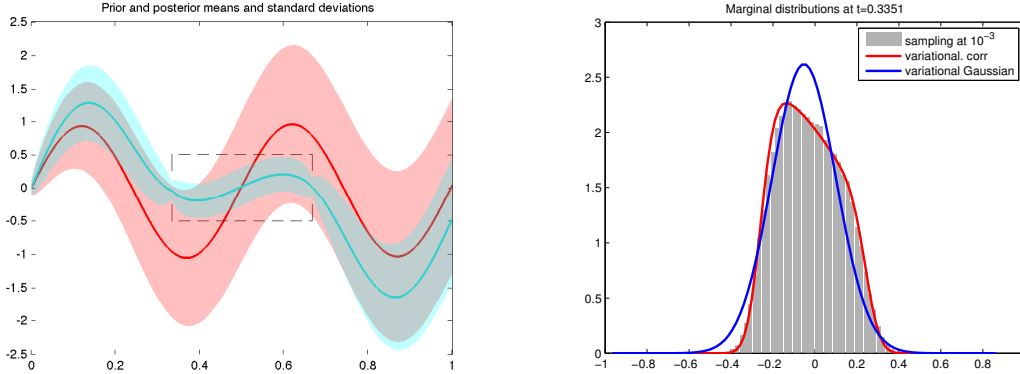

Figure 1: Inference results for the toy model in Section 3.1. The continuous time potential is defined as $V(t, x_t) = (2x_t)^8 I_{[1/2,2/3]}(t)$ and we assume two hard box discrete likelihood terms $I_{[-0.25,0.25]}(x_{t_1})$ and $I_{[-0.25,0.25]}(x_{t_2})$ placed at $t_1 = 1/3$ and $t_2 = 2/3$. The prior is defined by the parameters $a_t = -1$, $c_t = 4\pi \cos(4\pi t)$ and $b_t = 4$. The left panel shows the prior's and the posterior approximation's marginal means and standard deviations. The right panel shows the marginal approximations at $t = 0.3351$, a region where we expect the corrections to be strongly influenced by both types of likelihoods. Samples were generated by using the lag $\Delta t = 10^{-3}$, the approximate inference was run using RK4 at $\Delta t = 10^{-4}$.

## 3 Experiments

### 3.1 Inference in a (soft) box

The first example we consider is a mixed discrete-continuous time inference under box and soft box likelihood observations respectively. We consider a diffusing particle on the line under an OU prior process of the form

$$dx_t = (-ax_t + c_t)dt + \sqrt{b}dW_t$$

with $a = -1$, $c_t = 4\pi \cos(4\pi t)$ and $b = 4$. The likelihood model is given by the loss function $V(t, x_t) = (2x_t)^8$ for all $t \in [1/2, 2/3]$ and 0 otherwise, effectively confining the process to a narrow strip near zero (soft box). This likelihood is therefore an approximation to physically realistic situations where particles can perform diffusion in a confined environment. The box has hard gates: two discrete time likelihoods given by the indicator functions $I_{[-0.25,0.25]}(x_{t_1})$ and $I_{[-0.25,0.25]}(x_{t_2})$ placed at the ends of the interval, that is, $T_d = \{1/3, 2/3\}$. The left panel in Figure 1 shows the prior and approximate posterior processes (mean $\pm$ one standard deviation) in pink and cyan respectively: the confinement of the process to the box is in clear evidence, as well as the narrowing of the confidence intervals corresponding to the two discrete time observations. The right panel in Figure 1 shows the marginal approximations at a time point shortly after the "gate" to the box, these are: (i) sampling (grey) (ii) the Gaussian EP approximation (blue line), and (iii) its corrected version (red line). The time point was chosen as we expect the strongest non-Gaussian effects to be felt near the discrete likelihoods; the corrected distribution does indeed show strong skewness. To benchmark the method, we compare it to MCMC sampling obtained by using slice sampling [Murray et al., 2010] on the discretised model with $\Delta t = 10^{-3}$. We emphasise that this is an approximation to the model, hence the benchmark is not a true gold standard; however, we are not aware of sampling schemes that would be able to perform inference under the exact continuous time likelihood. The histogram in Figure 1 was generated from a sample size of $10^5$ following a burn in of $10^4$. The Gaussian EP approach gives a very good reconstruction of the first two moments of the distribution. The corrected EP approximation is very close to the MCMC results.

### 3.2 Log Gaussian Cox processes

Another family of models where one encounters continuous time likelihoods is point processes; these processes find wide application in a number of disciplines, from neuroscience Smith and Brown [2003] to conflict modelling Zammit-Mangion et al. [2012]. We assume that we have a multivariate log Gaussian Cox process model [Kingman, 1992]: this is defined by a $d$-variate Ornstein-Uhlenbeck process $\{\boldsymbol{x}_t\}_t$

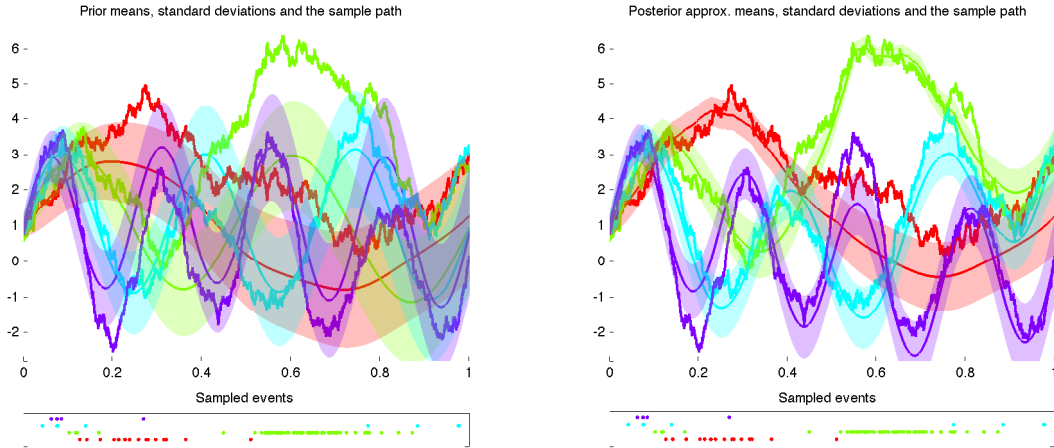

Figure 2: A toy example for the point process model in Section 3.2. The prior is defined by $\boldsymbol{A} = [-2, 1, 0, 1; 1, -2, 1, 0; 0, 1, -2, 1; 1, 0, 1, -2]$, $c_t^i = 4i\pi\cos(2i\pi t)$, $\boldsymbol{B} = 4\boldsymbol{I}$. We use $\mu_i = 0$. The prior means and standard deviations, the sampled process path, and the sampled events are shown on the left panel while the posterior approximations are shown on the right panel.

on the $[0, 1]$ interval. Conditioned on $\{\boldsymbol{x}_t\}_t$ we have $d$ Poisson point processes with intensities given by $\lambda_t^i = e^{\mu_i + x_t^i}$ for all $i = 1, \ldots, d$ and $t \in [0, 1]$. The likelihood of this point process model is formed by both discrete time (point probabilities) and continuous time (void probability) terms and can be written as

$$\log \prod_i p(\mathcal{Y}_i | \{x_t^i\}_t) \doteq \sum_i \Big\{ -e^{\mu_i} \int_0^1 dt e^{x_t^i} + |\mathcal{Y}_i|\mu_i + \sum_{t_k \in \mathcal{Y}_i} x_t^i \Big\},$$

where $\mathcal{Y}_i$ denotes the set of observed event times corresponding to $\{x_t^i\}_t$. Clearly, the discrete time observations in this model are (degenerate) Gaussians, therefore, one may opt for starting with an OU process with a translated drift, however, for consistency reasons, we treat them as discrete time observations.

In this example we chose $d = 4$ and $\boldsymbol{A} = [-2, 1, 0, 1; 1, -2, 1, 0; 0, 1, -2, 1; 1, 0, 1, -2]$, thus coupling the various processes. We chose $c_t^i = 4i\pi\cos(2i\pi t)$, $\boldsymbol{B} = 4\boldsymbol{I}$ and $\mu_i = 0$. We generate a sample path $\{\tilde{\boldsymbol{x}}_t\}_t$, draw observations $\mathcal{Y}_i$ based on $\{\tilde{x}_t^i\}_t$ and perform inference.

The results are shown in Figure 2, with four colours distinguishing the four processes. The left panel shows prior processes (mean $\pm$ standard deviation), sample paths and (bottom row) the sampled points (i.e. the data). The right panel shows the corresponding posterior processes approximations. The results reflect the general pattern characteristic of fitting point process data: in regions with a substantial number of events the sampled path can be inferred with great accuracy (accurate mean, low standard deviation) whereas in regions with no or only a few events the fit reverts to a skewed/shifted prior path, as the void probability dominates.

## 3.3 Point process modelling of neural spikes trains

In a third example we consider continuous time point process inference for spike time recordings from a population of neurons. This type of data is frequently modelled using (discrete time) state-space models with point process observations (SSPP) [Smith and Brown, 2003, Zammit Mangion et al., 2011, Macke et al., 2011]; parameter estimation from such models can reveal biologically relevant facts about the neuron's electrophysiology which are not apparent from the spike trains themselves. We consider a dataset from Di Lorenzo and Victor [2003], available at www.neurodatabase.org, consisting of recordings of spiking patterns of taste response cells in Sprague-Dawley rats during presentation of different taste stimuli. The recordings are $10s$ each at a resolution of $10^{-3}s$, and four different taste stimuli: (i) NaCL, (ii) Quinine HCl, (iii) Quinine HCl, and (iv) Sucrose are presented to the subjects for the duration of the first $5s$ of the $10s$ recording window. We modelled the spike train recordings by univariate log Gaussian Cox process models (see Section 3.2) with homogeneous OU priors, that is, $A_t, c_t$ and $B_t$ were considered constant. We use the variational EM algorithm (discrete time likelihoods are Gaussian) to learn the prior

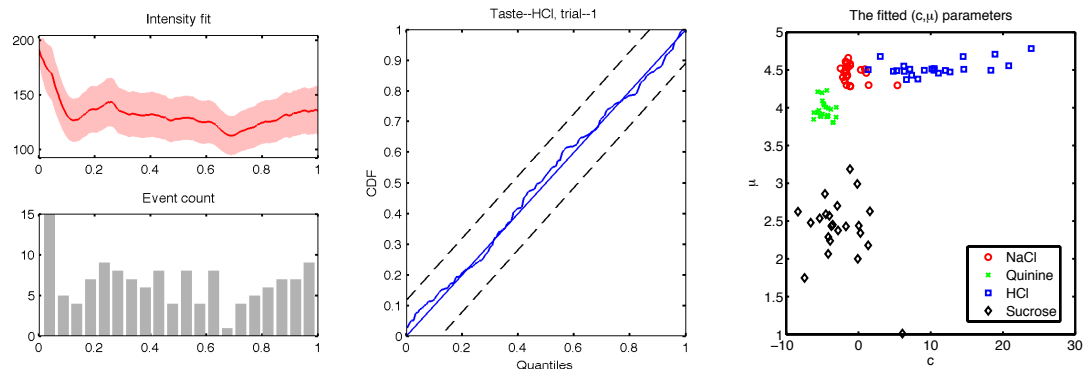

Figure 3: Inference results on data from cell 9 form the dataset in Section 3.3. The top-left, bottom-left and centre panels show the intensity fit, event count and the Q-Q plot corresponding to one of the recordings, whereas the right panel shows the learned $c$ and $\mu$ parameters for all spike trains in cell 9.

parameters $A$, $c$ and $\mu$ and initial conditions for each individual recording. We scaled the $10s$ window into the unit interval $[0, 1]$ and used a $10^{-4}$ resolution.

Fig 3 shows example results of this procedure. The right panel shows an emergent pattern of stimulus based clustering of $\mu$ and $c$ as in Zammit Mangion et al. [2011]. We observe that discrete-time approaches such as [Smith and Brown, 2003, Zammit Mangion et al., 2011] are usually forced to take very fine time discretisation by the requirement that at most one spike happens during one time step. This leads to significant computational resources being invested in regions with few spikes. Our continuous time approach, on the other hand, handles uneven observations naturally.

# 4 Conclusion

Inference methodologies for continuous time stochastic processes are a subject of intense research, both for fundamental and applied research. This paper contributes a novel approach which allows inference from both discrete time and continuous time observations. Our results show that the method is effective in accurately reconstructing marginal posterior distributions, and can be deployed effectively on real world problems. Furthermore, it has recently been shown [Kappen et al., 2012] that optimal control problems can be recast in inference terms: in many cases, the relevant inference problem is of the same type as the one considered here, hence this methodology could in principle also be used in control problems. The method is based on the parallel EP formulation of Cseke and Heskes [2011b]: interestingly, we show that the EP updates from continuous time observations collapse to variational updates [Archambeau et al., 2007]. Algorithmically, our approach results in efficient forward-backward updates, compared to the gradient ascent algorithm of Archambeau et al. [2007]. Furthermore, the EP perspective allows us to compute corrections to the Gaussian marginals; in our experiments, these turned out to be highly accurate.

Our modelling framework assumes a latent linear diffusion process; however, as mentioned before, some non-linear diffusion processes are equivalent to posterior processes for OU processes observed in continuous time [Øksendal, 2010]. Our approach, hence, can also be viewed as a method for accurate marginal computations in (a class of) nonlinear diffusion processes observed with noise. In general, all non-linear diffusion processes can be recast in a form similar to the one considered here; the important difference though is that the continuous time likelihood is in general an Ito integral, not a regular integral. In the future, it would be interesting to explore the extension of this approach to general non-linear diffusion processes, as well as discrete and hybrid stochastic processes [Rao and Teh, 2012, Ocone et al., 2013].

### Acknowledgements

B.Cs. is funded by BBSRC under grant BB/I004777/1. M.O. would like to thank for the support by EU grant FP7-ICT-270327 (Complacs). G.S. acknowledges support from the ERC under grant MLCS-306999.

## Footnotes

[1]We remark that one could also integrate the OU process between time steps, yielding an exact finite dimensional marginal of the prior. In the limit however both procedures are equivalent.

[2]We use "·" as scalar product for general (concatenated) vector objects, for example, $\boldsymbol{x} \cdot \boldsymbol{y} = \boldsymbol{x}^T \boldsymbol{y}$ when $\boldsymbol{x}, \boldsymbol{y} \in \mathbb{R}^n$.

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
