[Supplementary Material · ContObs-NIPS2013-SM.pdf]

# A  Supplementary Material

This text is the Supplementary Material of the paper *"Approximate inference in latent Gaussian-Markov models from continuous time observations"* by B. Cseke, M. Opper and G. Sanguinetti (*Neural Information Processing Systems* 2013).

## A.1  Variational formulation using expectation constraints

In this section we formulate an expectation constraints based approximate inference scheme [Heskes et al., 2005] for our model. It turns out that, when only discrete time observation are present, the inference results in an expectation propagation type algorithm whereas when only continuous type observations are present it collapses to the variational approach. These two approaches can be combined into a joint inference scheme.

### A.1.1  Discrete time observations

In the case of only discrete time observations no time discretisation is needed for the formal manipulation of the distributions. The propagation algorithm we arrive to can be viewed as an EP on a latent Gaussian model where the (partial) matrix inversion for computing marginal means and variances is replaced by solving the forward-backward differential equations.

In the following we present an expectation constrained free energy optimisation that leads to the EP algorithm. Here we use the concept of the variational formulation by free energies [e.g. Yedidia et al., 2000]. In a similar spirit as in Heskes et al. [2005], instead of approximating

$$p(\{\boldsymbol{x}_t\}|\{\boldsymbol{y}_{t_i}^d\}_i) \propto p_0(\{\boldsymbol{x}_t\}) \times \prod_i p(\boldsymbol{y}_{t_i}^d|\boldsymbol{x}_{t_i})$$

with an OU process we define the free energy as function of a family of approximate marginals $\mathcal{Q} = \{q_0(\{\boldsymbol{x}_t\}), \{q_{ds}^i(\boldsymbol{x}_{t_i})\}_i, \{q_d^i(\boldsymbol{x}_{t_i})\}_i\}$ constrained by expectation constraints. The densities $q_i^d$ will be assigned to the factors corresponding to the likelihood terms, $q_0$ will assigned to the factor $p_0$. With some abuse in notation the family $\mathcal{Q}$ can be viewed as representing an approximation $q$ having the form

$$q(\{\boldsymbol{x}_t\}) \propto \frac{q_0(\{\boldsymbol{x}_t\}) \prod_i q_d^i(\boldsymbol{x}_{t_i})}{\prod_i q_{ds}^i(\boldsymbol{x}_{t_i})}.$$

Note that in the graphical model formalism the densities $q_{ds}^i$ correspond to the densities defined over the variables of the separator sets in graphical models [Lauritzen, 1996]. The expectation constraints are defined over the function $\boldsymbol{f}(\boldsymbol{z}) = (\boldsymbol{z}, -\boldsymbol{z}\boldsymbol{z}^T/2)$ and ensure that the corresponding marginals of the members of $\mathcal{Q}$ are consistent up to second moments, i.e., their marginal means and covariances are equal. Given the above assumptions, one can define an approximation of the $\mathrm{D}[q\|p]$ divergence called free energy that reads as

$$
\begin{aligned}
F(\mathcal{Q}) = &- \langle \log p_0(\{\boldsymbol{x}_t\}) \rangle_{q_0} - \sum_i \left\langle \log p(\boldsymbol{y}_i^d|\boldsymbol{x}_t) \right\rangle_{q_d^i} \\
&+ \langle \log q_0(\{\boldsymbol{x}_t\}) \rangle_{q_0} + \sum_i [\left\langle \log q_d^i(\boldsymbol{x}_{t_i}) \right\rangle_{q_d^i} - \left\langle \log q_{ds}^i(\boldsymbol{x}_{t_i}) \right\rangle_{q_{ds}^i}]
\end{aligned}
\tag{16}
$$

and specify the expectation constraints

$$\langle \boldsymbol{f}(\boldsymbol{x}_{t_i}) \rangle_{q_0} = \langle \boldsymbol{f}(\boldsymbol{x}_{t_i}) \rangle_{q_{ds}^i} \quad \text{and} \quad \langle \boldsymbol{f}(\boldsymbol{x}_{t_i}) \rangle_{q_d^i} = \langle \boldsymbol{f}(\boldsymbol{x}_{t_i}) \rangle_{q_{ds}^i} \quad \text{for all } t_i \in T_d.$$

The stationary equations of the corresponding Lagrangian

$$L(\mathcal{Q}, \Lambda) = F(\mathcal{Q}) + \sum_i \left[ \boldsymbol{\lambda}_{t_i}^0 \cdot [\langle \boldsymbol{f}(\boldsymbol{x}_{t_i}) \rangle_{q_{ds}^i} - \langle \boldsymbol{f}(\boldsymbol{x}_{t_i}) \rangle_{q_0}] + \boldsymbol{\lambda}_{t_i}^d \cdot [\langle \boldsymbol{f}(\boldsymbol{x}_{t_i}) \rangle_{q_{ds}^i} - \langle \boldsymbol{f}(\boldsymbol{x}_{t_i}) \rangle_{q_d^i}] \right] \tag{17}$$

result in $q_0(\{\boldsymbol{x}_t\}) \propto p_0(\{\boldsymbol{x}_t\}) \times \exp(\sum_i \boldsymbol{\lambda}_{t_i}^0 \cdot \boldsymbol{f}(\boldsymbol{x}_{t_i}))$, $q_d^i(\boldsymbol{x}_{t_i}) \propto p(\boldsymbol{y}_i^d|\boldsymbol{x}_{t_i}) \times \exp(\boldsymbol{\lambda}_{t_i}^d \cdot \boldsymbol{f}(\boldsymbol{x}_{t_i}))$ and $q_{ds}^i(\boldsymbol{x}_{t_i}) \propto \exp([\boldsymbol{\lambda}_{t_i}^0 + \boldsymbol{\lambda}_{t_i}^d] \cdot \boldsymbol{f}(\boldsymbol{x}_{t_i}))$. The differential w.r.t. the Lagrange multipliers $\boldsymbol{\lambda}_{t_i}^0$ and $\boldsymbol{\lambda}_{t_i}^d$ lead to the above mentioned expectation constraints. Since the expectation constraints are defined over the sufficient statistics $\boldsymbol{f}(\boldsymbol{z})$ and the optimal $q_{ds}^i$ belongs to the exponential (Gaussian) family defined by $\boldsymbol{f}$, we can rewrite these constraints into canonical forms. These read as

$$\boldsymbol{\lambda}_{t_i}^d + \boldsymbol{\lambda}_{t_i}^0 = \mathrm{Collapse}(q_0(\boldsymbol{x}_{t_i}); \boldsymbol{f}) \quad \text{and} \quad \boldsymbol{\lambda}_{t_i}^d + \boldsymbol{\lambda}_{t_i}^0 = \mathrm{Collapse}(q_d^i(\boldsymbol{x}_{t_i}); \boldsymbol{f}).$$

Here Collapse$(q(\boldsymbol{z}); \boldsymbol{f})$ denotes the (unique) moment matching canonical parameters, in other words the Kullback-Leibler projection Collapse$(q(\boldsymbol{z}); \boldsymbol{f}) = \operatorname{argmin}_{\boldsymbol{\theta}} \mathrm{D}[q(\boldsymbol{z})||\exp(\boldsymbol{\theta} \cdot \boldsymbol{f}(\boldsymbol{z}) - \log Z(\boldsymbol{\theta}))]$. From the moment matching constraints one can introduce the fixed point iteration

$$[\boldsymbol{\lambda}_{t_i}^0]^{new} = \text{Collapse}(q_d^i(\boldsymbol{x}_{t_i}); \boldsymbol{f}) - \boldsymbol{\lambda}_{t_i}^d, \tag{18}$$

$$[\boldsymbol{\lambda}_{t_i}^d]^{new} = \text{Collapse}(q_0(\boldsymbol{x}_{t_i}); \boldsymbol{f}) - \boldsymbol{\lambda}_{t_i}^0, \tag{19}$$

which corresponds to a (parallel) EP algorithm in a latent Gaussian model where the latent Gaussian is given by an OU process. It can be shown that the free energy in (16) is finite, details are given in Section A.2. The collapse operation Collapse$(q_d^i(\boldsymbol{x}_{t_i}); \boldsymbol{f})$ is computed by moment matching—computing the corresponding canonical parameters—while Collapse$(q_0(\boldsymbol{x}_{t_i}); \boldsymbol{f})$ is computed by using the moment parameters of the marginals resulting from the forward-backward equations in Section B.1, with $\boldsymbol{\lambda}_{t_i}^0 = (\boldsymbol{h}_{t_i}^d, \boldsymbol{Q}_{t_i}^d)$. Readers familiar with the EP presented in Opper and Winther [2000] or [Minka, 2001] can identify the multipliers $\boldsymbol{\lambda}_{t_i}^0$ as the canonical parameters of the so called term approximations whereas the $\boldsymbol{\lambda}_{t_i}^d$s correspond to the canonical parameters of the so called cavity distributions. Equations (18) and (19) correspond to the updates of the term approximation and the cavity distribution through moment matching. The free energy approach presented above starts from the variational formulation $\mathrm{D}[q||p]$ where instead of single specially chosen Gaussian $q$, a family of approximate marginals $\mathcal{Q}$ is introduced. The EP style iterative moment matching minimizations in the $\mathrm{D}[p||\cdot]$ sense corresponding to Collapse$(\cdot; \boldsymbol{f})$, arise from the satisfaction of moment matching constraints.

Clearly, any method that computes the marginal means and covariances of $q_0$ at the time-point in $T_d$ suffices to keep the iteration running, and thus, when possible, one should solve the differential equations between observation points analytically. We can also opt for the alternative generic approach of computing the covariance matrix corresponding to the variables $\{\boldsymbol{x}_{t_i}\}_{t_i \in T_d}$ and opt for the equivalent Gaussian process (OU covariance function) expectation propagation in Opper and Winther [2000] or Minka [2001]. In the latter case the marginal means and variances for $t \notin T_d$ can be computed by using the conditional independencies in the model and computing the predictive distributions.

### A.1.2 Continuous time observations

In this section we extend the approach to the case when only continuous tine observations are present. The task is to approximate a posterior distribution having the form

$$p(\{\boldsymbol{x}_t\}|\{\boldsymbol{y}_t^c\}) \propto p(\{\boldsymbol{x}_t\}) \times \exp\left\{-\int_0^1 dt V(t, \boldsymbol{y}_t^c, \boldsymbol{x}_t)\right\}.$$

In order to simplify notation, we will omit the dependence of $V$ on $\boldsymbol{y}_t^c$. In an similar fashion as in the previous section we introduce a family of marginals $\mathcal{Q} = \{q_0(\{\boldsymbol{x}_t\}), q_{cs}(\{\boldsymbol{x}_t\}), q_c(\{\boldsymbol{x}_t\})\}$ and define the free energy as

$$F(\mathcal{Q}) = -\langle\log p_0(\{\boldsymbol{x}_t\})\rangle_{q_0} + \left\langle\int_0^1 dt V(t, \boldsymbol{x}_t)\right\rangle_{q_c} + \langle\log q_0(\{\boldsymbol{x}_t\})\rangle_{q_0} + \langle\log q_c(\{\boldsymbol{x}_t\})\rangle_{q_c} - \langle\log q_{cs}(\{\boldsymbol{x}_t\})\rangle_{q_{cs}} \tag{20}$$

The moment matching constraints will be defined as

$$\langle\boldsymbol{f}(\boldsymbol{x}_t)\rangle_{q_0} = \langle\boldsymbol{f}(\boldsymbol{x}_t)\rangle_{q_{cs}} \quad \text{and} \quad \langle\boldsymbol{f}(\boldsymbol{x}_t)\rangle_{q_c} = \langle\boldsymbol{f}(\boldsymbol{x}_t)\rangle_{q_{cs}} \quad \text{for all } t \in [0, 1].$$

which imply using Lagrange multiplier terms of the form

$$C(\mathcal{Q}, \mathcal{M}) = \int_0^1 dt\, \boldsymbol{\mu}_t^0 \cdot [\langle\boldsymbol{f}(\boldsymbol{x}_t)\rangle_{q_{cs}} - \langle\boldsymbol{f}(\boldsymbol{x}_t)\rangle_{q_0}] + \int_0^1 dt\, \boldsymbol{\mu}_t^c \cdot [\langle\boldsymbol{f}(\boldsymbol{x}_t)\rangle_{q_{cs}} - \langle\boldsymbol{f}(\boldsymbol{x}_t)\rangle_{q_c}]. \tag{21}$$

In order to carry out the computations and show that the above quantities exist, we discretise the time domain by using the time-points $T = \{t_0 = 0, t_1, \ldots, t_{K-1}, t_K = 1\}$ with the lags $\Delta t_k = t_{k+1} - t_k$ and represent the process $\{\boldsymbol{x}_t\}_t$ by the matrix $\boldsymbol{x} = [\boldsymbol{x}_{t_0}, \ldots, \boldsymbol{x}_{t_K}]$. By using this discretisation, we approximate all integrals using the corresponding Euler discretisation and,, view $p_0(\boldsymbol{x})$ as a multivariate Gaussian. We use the indexing $T$ to highlight the discretisation. We define the Lagrangian as

$$L(\mathcal{Q}_T, \mathcal{M}_T) = F(\mathcal{Q}_T) + C(\mathcal{Q}_T, \mathcal{M}_T). \tag{22}$$

The stationary conditions (22) corresponding to the differentiation w.r.t $q_0, q_{cs}$ and $q_c$, result in

$$q_0(\boldsymbol{x}) \propto p_0(\boldsymbol{x}) \times \exp\left\{\sum_k \Delta t_k \boldsymbol{\mu}_{t_k}^0 \cdot \boldsymbol{f}(\boldsymbol{x}_{t_k})\right\}, \tag{23}$$

$$q_c(\boldsymbol{x}) \propto \exp\left\{\sum_k \Delta t_k[-V(t_k, \boldsymbol{x}_{t_k}) + \boldsymbol{\mu}_{t_k}^c \cdot \boldsymbol{f}(\boldsymbol{x}_{t_k})]\right\}, \tag{24}$$

$$q_{cs}(\boldsymbol{x}) \propto \exp\left\{\sum_k \Delta t_k[\boldsymbol{\mu}_{t_k}^0 + \boldsymbol{\mu}_{t_k}^c] \cdot \boldsymbol{f}(\boldsymbol{x}_{t_k})\right\}. \tag{25}$$

Due to the factorisation of $q_{cs}$ and $q_c$ and the Gaussian nature of $q_{cs}(\boldsymbol{x}_{t_k})$, the stationary conditions corresponding to the moment constraints can be rewritten as

$$\Delta t_k[\boldsymbol{\mu}_{t_k}^0 + \boldsymbol{\mu}_{t_k}^c] = \text{Collapse}(q_0(\boldsymbol{x}_{t_k}); \boldsymbol{f}) \quad \text{and} \quad \Delta t_k[\boldsymbol{\mu}_{t_k}^0 + \boldsymbol{\mu}_{t_k}^c] = \text{Collapse}(q_c(\boldsymbol{x}_{t_k}); \boldsymbol{f}) \quad \text{for all, } t_k \in T. \tag{26}$$

Taking the limit $\Delta t_k \to 0$ is not feasible at this point because the marginals of both $q_{cs}$ and $q_c$ collapse into delta distributions. However, we can observe that $\text{Collapse}(q_0(\boldsymbol{x}_{t_k}); \boldsymbol{f})$ should always be finite and well defined. We use the alias $\boldsymbol{\mu}_{t_k} = \text{Collapse}(q_0(\boldsymbol{x}_{t_k}; \boldsymbol{f})$ and we eliminate $\boldsymbol{\mu}_{t_k}^c$ from the formulae above. In an similar spirit as in Section A.1.1, we use the moment matching constraint to define the fixed point iteration

$$[\boldsymbol{\mu}_{t_k}^0]^{new} = \boldsymbol{\mu}_{t_k}^0 + \frac{1}{\Delta t_k}[\text{Collapse}(q_c(\boldsymbol{x}_{t_k}); \boldsymbol{f}) - \boldsymbol{\mu}_{t_k}], \tag{27}$$

where, due to eliminating $\boldsymbol{\mu}_{t_k}^c$, we have $q_c(\boldsymbol{x}_{t_k}) \propto \exp\{-\Delta t_k[V(t_k, \boldsymbol{x}_{t_k}) + \boldsymbol{\mu}_{t_k}^0 \cdot \boldsymbol{f}(\boldsymbol{x}_{t_k})] + \boldsymbol{\mu}_{t_k} \cdot \boldsymbol{f}(\boldsymbol{x}_{t_k})\}$. The $\Delta t_k \to 0$ limit is presented in Section 2.2.3 of the paper and by using $\boldsymbol{\mu}_t^0 = (\boldsymbol{h}_t^c, \boldsymbol{Q}_t^c)$, it results in the updates

$$[\boldsymbol{h}_t^c]^{new} = -\partial_{\boldsymbol{m}_t} \langle V(t, \boldsymbol{x}_t)\rangle_{q_0(\boldsymbol{x}_t)} + 2\partial_{\boldsymbol{V}_t} \langle V(t, \boldsymbol{x}_t)\rangle_{q_0(\boldsymbol{x}_t)} \boldsymbol{m}_t \quad \text{and} \quad [\boldsymbol{Q}_t^c]^{new} = \partial_{\boldsymbol{V}_t} \langle V(t, \boldsymbol{x}_t)\rangle_{q_0(\boldsymbol{x}_t)} \tag{28}$$

$$[q_0(\{\boldsymbol{x}_t\})]^{new} \propto p_0(\{\boldsymbol{x}_t\}) \times \exp\{\int_0^1 dt[\boldsymbol{x}_t^T \boldsymbol{h}_t^c - \frac{1}{2}\boldsymbol{x}_t^T \boldsymbol{Q}_t^c \boldsymbol{x}_t]\}$$

where the marginal moments of $q_0(\{\boldsymbol{x}_t\})$ are computed by using the Kalman-Bucy algorithm (Section B.1).

### A.1.3 The joint approximation scheme

Now that we have derived the approximation scheme for both discrete and continuous time observations, we can show that they can be easily combined to obtain a joint approximation. Without loss of generality, we can assume that $T_d \subset T$. By defining the joint family as $\mathcal{Q} = \{q_0(\boldsymbol{x}), \{q_d^i(\boldsymbol{x}_{t_i})\}_i, \{q_{ds}^i(\boldsymbol{x}_{t_i})\}_i, q_c(\boldsymbol{x})q_{cs}(\boldsymbol{x}), q_c(\boldsymbol{x})\}$ and the free energy as

$$F(\mathcal{Q}) = -\langle \log p_0(\boldsymbol{x})\rangle_{q_0} - \sum_i \left\langle \log p(\boldsymbol{y}_{t_i}^d | \boldsymbol{x}_t)\right\rangle_{q_d^i} + \left\langle \int_0^1 dt V(t, \boldsymbol{x}_t)\right\rangle_{q_c} \tag{29}$$

$$\langle \log q_0(\boldsymbol{x})\rangle_{q_0} + \sum_i [\left\langle \log q_d^i(\boldsymbol{x}_{t_i})\right\rangle_{q_d^i} - \left\langle \log q_{ds}^i(\boldsymbol{x}_{t_i})\right\rangle_{q_{ds}^i}] + \langle \log q_c(\boldsymbol{x})\rangle_{q_c} - \langle \log q_{cs}(\boldsymbol{x})\rangle_{q_{cs}}$$

we can construct the Lagrangian by using the multiplier terms from (17) and the Euler discretisation of (21). The fixed point iteration follows (18), (19) and (28) where $q_0$ is defined by

$$[q_0(\{\boldsymbol{x}_t\})]^{new} \propto p_0(\{\boldsymbol{x}_t\}) \times \exp\left\{\sum_i [\boldsymbol{x}_{t_i}^T \boldsymbol{h}_{t_i}^d - \frac{1}{2}\boldsymbol{x}_{t_i}^T \boldsymbol{Q}_{t_i}^d \boldsymbol{x}_{t_i}] + \int_0^1 dt[\boldsymbol{x}_t^T \boldsymbol{h}_t^c - \frac{1}{2}\boldsymbol{x}_t^T \boldsymbol{Q}_t^c \boldsymbol{x}_t]\right\}. \tag{30}$$

We use the forward-backward equations (Section B) to compute the marginals of $q_0$. As a result we have an algorithm behaves like a EP/variational hybrid: the parameters corresponding to the discrete time observations follow an EP style update (18) and (19), while the ones corresponding to the continuous observations are updated in a variational fashion according to (28).

### A.2 The computation of the free energy

In the following we show that the free energy exist when $\Delta t_k \to 0$.

### A.2.1 Continuous time observations

The expression of the free energy in (20) after discretisation is

$$F(\mathcal{Q}_T) = -\langle \log p_0(\boldsymbol{x}) \rangle_{q_0} + \sum_k \Delta t_k \langle V(t_k, \boldsymbol{x}_{t_k}) \rangle_{q_c} + \langle \log q_0(\boldsymbol{x}) \rangle_{q_0} + \langle \log q_c(\boldsymbol{x}) \rangle_{q_c} - \langle \log q_{cs}(\boldsymbol{x}) \rangle_{q_{cs}}$$

by substituting (23), (24) and (25) into $F(\mathcal{Q}_T)$ we find that

$$F(\mathcal{Q}_T) = -\log Z_0(\{\boldsymbol{\mu}_{t_k}^0\}) - \log Z_c(\{\boldsymbol{\mu}_{t_k}^c\}) + \log Z_{cs}(\{\boldsymbol{\mu}_{t_k}^0 + \boldsymbol{\mu}_{t_k}^c\})$$

where $Z_0$, $Z_c$ and $Z_{cs}$ stand for the corresponding normalisation constants. By using the Legendre duality we can write

$$-\log Z_0(\{\boldsymbol{\mu}_{t_k}^0\}) = \mathrm{D}[q_0(\boldsymbol{x})||p_0(\boldsymbol{x})] - \sum_k \Delta t_k \boldsymbol{\mu}_{t_k}^0 \cdot \boldsymbol{f}(\boldsymbol{x}_{t_k})$$

and by using the expansion in (11) in Section 2.2.3 of the paper, we find that

$$-\log Z_c(\{\boldsymbol{\mu}_{t_k}^c\}) + \log Z_{cs}(\{\boldsymbol{\mu}_{t_k}^0 + \boldsymbol{\mu}_{t_k}^c\}) \simeq \sum_k \Delta t_k \langle V(t_k, \boldsymbol{x}_{t_k}) + \boldsymbol{\mu}_{t_k}^0 \cdot \boldsymbol{f}(\boldsymbol{x}_{t_k}) \rangle_{q_0}.$$

Since $q_0$ corresponds to an OU process (see Section B.3 for its parametric form), we can take the limit $\Delta t_k \to 0$ and obtain

$$\begin{aligned}
F(\mathcal{Q}) =& \mathrm{D}[q_0(\{\boldsymbol{x}_t\})||p_0(\{\boldsymbol{x}_t\})] + \int_0^1 dt \, \langle V(t, \boldsymbol{x}_t) \rangle_{q_0} \\
=& \frac{1}{2} \int_0^1 dt \, \left\langle [(\boldsymbol{A}_t - \boldsymbol{A}_t^q)\boldsymbol{x}_t + (\boldsymbol{c}_t - \boldsymbol{c}_t^q)]^T \boldsymbol{B}_t^{-1} [(\boldsymbol{A}_t - \boldsymbol{A}_t^q)\boldsymbol{x}_t + (\boldsymbol{c}_t - \boldsymbol{c}_t^q)] \right\rangle_{q_0} \\
&+ \mathrm{D}[p_0(\boldsymbol{x}_0)||q_0(\boldsymbol{x}_0)] + \int_0^1 dt \, \langle V(t, \boldsymbol{x}_t) \rangle_{q_0},
\end{aligned}$$

where $\boldsymbol{A}_t^q$ and $\boldsymbol{c}_t^q$ represent the parameters corresponding to $q_0$. Computing $\mathrm{D}[q_0(\{\boldsymbol{x}_t\})||p_0(\{\boldsymbol{x}_t\})]$ when $q_0$ and $p_0$ are parameterised OU processes can be done as in [e.g. Archambeau et al., 2007].

### A.2.2 Discrete time observations

We use the notation

$$q_d^i(\boldsymbol{x}_{t_i}) = \frac{1}{Z_d^i} p(\boldsymbol{y}_i^d|\boldsymbol{x}_{t_i}) \times \exp(\boldsymbol{\lambda}_i^d \cdot \boldsymbol{f}(\boldsymbol{x}_{t_i})),$$

$$q_{ds}^i(\boldsymbol{x}_{t_i}) = \frac{1}{Z_{ds}^i} \exp([\boldsymbol{\lambda}_i^0 + \boldsymbol{\lambda}_i^d] \cdot \boldsymbol{f}(\boldsymbol{x}_{t_i}))$$

and by using the Legendre duality as above, we can rewrite (16) as

$$F(\mathcal{Q}) = \mathrm{D}[q_0(\{\boldsymbol{x}_t\})||p_0(\{\boldsymbol{x}_t\})] - \sum_i \boldsymbol{\lambda}_i^0 \cdot \langle \boldsymbol{f}(\boldsymbol{x}_{t_i}) \rangle_{q_{ds}^i} - \sum_i [\log Z_d^i - \log Z_{ds}^i],$$

which is a finite, computable quantity. The joint free energy follows from combining the discrete and continuous free energies according to (29), that is,

$$\begin{aligned}
F(\mathcal{Q}) =& \mathrm{D}[p_0(\boldsymbol{x}_0)||q_0(\boldsymbol{x}_0)] + \frac{1}{2} \int_0^1 dt \, \left\langle [(\boldsymbol{A}_t - \boldsymbol{A}_t^q)\boldsymbol{x}_t + (\boldsymbol{c}_t - \boldsymbol{c}_t^q)]^T \boldsymbol{B}_t^{-1} [(\boldsymbol{A}_t - \boldsymbol{A}_t^q)\boldsymbol{x}_t + (\boldsymbol{c}_t - \boldsymbol{c}_t^q)] \right\rangle_{q_0} \\
&+ \int_0^1 dt \, \langle V(t, \boldsymbol{x}_t) \rangle_{q_0} - \sum_i \boldsymbol{\lambda}_i^0 \cdot \langle \boldsymbol{f}(\boldsymbol{x}_{t_i}) \rangle_{q_{ds}^i} - \sum_i [\log Z_d^i - \log Z_{ds}^i].
\end{aligned}$$

Note that after convergence we have $q_{ds}^i(\boldsymbol{x}_t) = q_0(\boldsymbol{x}_t)$.

## B   Computations related to the Kalman-Bucy forward-backward algorithm

This section contains the computations that complement the material presented in Section 2.1. For reasons of simplicity, in Sections B.2 and B.3 we focus on the continuos time case, the computations related to the additional discrete time terms follow naturally.

## B.1 The Kalman-Bucy forward-backward equations

By using the Euler discretisation and first order expansions as in [e.g. Särkkä, 2006] one can show that the forward and backward filtering equations satisfy

$$\frac{d}{dt}V_t^{fw} = A_t V_t^{fw} + V_t^{fw} A_t^T + B_t - V_t^{fw} Q_t^c V_t^{fw}, \qquad m_{t_i+}^{fw} = (I + V_{t_i}^{fw} Q_{t_i}^d)^{-1}(m_{t_i}^{fw} + V_{t_i}^{fw} h_{t_i}^d),$$

$$\frac{d}{dt}m_t^{fw} = A_t m_t^{fw} + c_t + V_t^{fw}[h_t^c - Q_t^c m_t^{fw}], \qquad V_{t_i+}^{fw} = (I + V_t^{fw} Q_{t_i}^d)^{-1} V_t^{fw},$$

$$\frac{d}{dt}V_t^{bw} = A_t V_t^{bw} + V_t^{bw} A_t^T - B_t + V_t^{bw} Q_t^c V_t^{bw}, \qquad m_{t_i-}^{bw} = (I + V_{t_i}^{bw} Q_{t_i}^d)^{-1}(m_{t_i}^{bw} + V_{t_i}^{bw} h_{t_i}^d),$$

$$\frac{d}{dt}m_t^{bw} = A_t m_t^{bw} + c_t - V_t^{bw}[h_t^c - Q_t^c m_t^{bw}], \qquad V_{t_i-}^{bw} = (I + V_{t_i}^{bw} Q_{t_i}^d)^{-1} V_{t_i}^{bw}.$$

We solve the equations for $m^{fw}$ and $V^{fw}$ in a forward fashion using the initial conditions $N(x_0; m_0, V_0)$, whereas the equations for $m^{fw}$ and $V^{fw}$ are solved in a backwards with the initial, or more specifically, the end conditions given by the a non-informative Gaussian. By combining the forward and backward solutions we obtain the posterior marginal density $p(x_t|\{y_{t_i}^d\}_i, \{y_t^c\}) \propto N(x_t; m_t^{fw}, V_t^{fw}) \times N(x_t; m_t^{bw}, V_t^{bw})$. Note that the backward equations are often replaced by the so called smoothing equations [e.g. Särkkä, 2006]. Thsese combine the backward equations and the latter computation of the marginals into a pair of differential equations for the mean and the covariance respectively. In some cases one is better off with computing directly the inverse of $V_t^{fw}$ or $V_t^{bw}$, these also follow similar quadratic or linear differential equations as the ones above.

## B.2 The variational approach to the Kalman-Bucy problem

In this section we present the computations of $\mathrm{D}[q(\{x_t\})||p(\{x_t\}|\{y_t^c\})]$ for the probabilistic model corresponding to the Kalman-Bucy problem defined by the equations

$$dx_t = (A_t x_t + c_t)dt + B_t^{1/2} dW_t, \quad \text{and} \quad dy_t = H_t x_t dt + R_t^{1/2} dW_t.$$

We relate its optimum's marginals to the marginals computed by the Kalman-Bucy algorithm. Let us discretise (again) by using the Euler scheme and write

$$p(x|y^c) \propto N(x_0; m_0, V_0)) \prod_k N(x_{t_{k+1}}; x_{t_k} + (A_{t_k} x_{t_k} + c_{t_k})\Delta t_k, \Delta t_k B_{t_k})$$

$$\times \prod_k N(y_{t_{k+1}}^c; y_{t_k}^c + H_{t_k} x_{t_k} \Delta t_k, \Delta t_k R_{t_k})$$

and assume that we approximate this density by a $q(x)$ which is the discretisation

$$q(x) \propto N(x_0; m_0, V_0)) \prod_k N(x_{t_{k+1}}; x_{t_k} + (A_{t_k}^q x_{t_k} + c_{t_k}^q)\Delta t_k, \Delta t_k B_{t_k})$$

of an approximating $dx_t = (A_t^q x_t + c_t^q)dt + B_t^{1/2} dW_t$, say, with same initial conditions. After some algebra, one can show that

$$\mathrm{D}[q(x)||p(x|y)] = \frac{1}{2}\sum_k \Delta t_k \left\langle [(A_{t_k} - A_{t_k}^q)x_{t_k} + (c_t - c_{t_k}^q)]^T B_{t_k}^{-1}[(A_{t_k} - A_{t_k}^q)x_{t_k} + (c_{t_k} - c_{t_k}^q)] \right\rangle_{q(x_{t_k})}$$

$$+ \frac{1}{2}\sum_k \Delta t_k \left\langle \left[\frac{\Delta y_{t_k}^c}{\Delta t_k} - H_{t_k} x_{t_k}\right]^T R_{t_k}^{-1}\left[\frac{\Delta y_{t_k}^c}{\Delta t_k} - H_{t_k} x_{t_k}\right] \right\rangle_{q(x_{t_k})} + \frac{d}{2}\sum_k \log(2\pi \Delta t_k R_{t_k}).$$

Clearly, due to the $\sum_k \log(2\pi \Delta t_k R_{t_k})$ terms, the limit $\Delta t_k \to 0$ does not exist but since these are not dependent on the variational parameters, one can still define the free energy

$$F(q) = \frac{1}{2}\int_0^1 dt \left\langle [(A_t - A_t^q)x_t + (c_t - c_t^q)]^T B_t^{-1}[(A_t - A_t^q)x_t + (c_t - c_t^q)] \right\rangle_{q(x_t)}$$

$$+ \frac{1}{2}\int_0^1 dt \left\langle \left[\frac{dy_t^c}{dt} - H_t x_t\right]^T R_t^{-1}\left[\frac{dy_t^c}{dt} - H_t x_t\right] \right\rangle_{q(x_t)}.$$

Due to the Gaussian nature of the problem, the Kalman-Bucy algorithm provides the marginal means and variances of the optimal $q$ corresponding to a quadratic loss function. As we show in Section B.3 below, the Kalman-Bucy algorithm can also be used to compute the $A_t^q$ and $c_t^q$ parameters of the optimal $q$. The case with additional discrete observations follows naturally.

### B.3 The moment matching OU to a Kalman-Bucy solution

Suppose now that we have computed, by Kalman-Bucy smoothing, the marginals of a process $q$. Since both the prior and observation processes are linear, the posterior process will also be of OU type; however, Kalman-Bucy smoothing only computes marginals of the process, and it may be expedient to compute the SDE formulation of the process. To do that, one needs the drift coefficients $\boldsymbol{A}_t$, $\boldsymbol{c}_t$ and the diffusion matrix $\boldsymbol{B}_t$. In order to compute these parameters, we resort to a variational computation.

We consider the (discretised) KL divergence between a posterior process $q(\boldsymbol{x})$ arising from a Kalman-Bucy problem and an OU process $p(\boldsymbol{x})$ with parameters $\boldsymbol{A}_t$, $\boldsymbol{c}_t$ and $\boldsymbol{B}_t$

$$\mathrm{D}[q(\boldsymbol{x})||p(\boldsymbol{x})] = \text{const.} + \langle \log q(\boldsymbol{x}) \rangle_q$$
$$+ \frac{1}{2} \sum_t \left\langle [\Delta \boldsymbol{x}_t - (\boldsymbol{A}_t \boldsymbol{x}_t + \boldsymbol{c}_t)\Delta t][\Delta t \boldsymbol{B}_t]^{-1}[\Delta \boldsymbol{x}_t - (\boldsymbol{A}_t \boldsymbol{x}_t + \boldsymbol{c}_t)\Delta t]^T + \log \det(\Delta t \boldsymbol{B}_t) \right\rangle_q$$

To find the optimal $\boldsymbol{A}_t^*, \boldsymbol{c}_t^*$ and $\boldsymbol{B}_t^*$ we need to compute the expectations needed in the above expression. Let us denote the forward filtering distribution at time $t$ by $N(\boldsymbol{x}_t; \boldsymbol{m}_t^{fw}, \boldsymbol{V}_t^{fw})$ while the backward filtering at $t+\Delta t$ is represented by $N(\boldsymbol{x}_{t+\Delta t}; \boldsymbol{m}_{t+\Delta t}^{bw}, \boldsymbol{V}_{t+\Delta t}^{bw})$; these distributions are known as they are the outcome of the Kalman-Bucy forward-backward filtering. The joint posterior of $(\boldsymbol{x}_t, \boldsymbol{x}_{t+\Delta t})$ can then be written as

$$q(\boldsymbol{x}_t, \boldsymbol{x}_{t+\Delta t}) \propto N(\boldsymbol{x}_t; \boldsymbol{m}_t^{fw}, \boldsymbol{V}_t^{fw}) N(\boldsymbol{y}_{t+\Delta t}^c; \boldsymbol{y}_c + \boldsymbol{H}_t \boldsymbol{x}_t \Delta t, \Delta t \boldsymbol{R}_t)$$
$$\times N(\boldsymbol{x}_{t+\Delta t}; (\boldsymbol{I} + \boldsymbol{A}_t \Delta t)\boldsymbol{x}_t + \boldsymbol{c}_t \Delta t, \Delta t \boldsymbol{B}_t) N(\boldsymbol{x}_{t+\Delta t}; \boldsymbol{m}_{t+\Delta t}^{bw}, \boldsymbol{V}_{t+dt}^{bw}).$$

This density can be rewritten in a conditional form

$$q(\boldsymbol{x}_t, \boldsymbol{x}_{t+\Delta t}) \propto N(\boldsymbol{x}_{t+\Delta t}; \boldsymbol{U}_t \boldsymbol{x}_t + \boldsymbol{v}_t \Delta t, \Delta t \boldsymbol{Z}_t) N(\boldsymbol{x}_t; \hat{\boldsymbol{m}}_t, \hat{\boldsymbol{V}}_t).$$

where the first order approximations of the quantities above are given by

$$\hat{\boldsymbol{m}}_t = \boldsymbol{m}_t^{fw} + \Delta t \boldsymbol{V}_t^{fw} \boldsymbol{H}_t^T (\Delta t \boldsymbol{H}_t \boldsymbol{V}_t^{fw} \boldsymbol{H}_t^T + \boldsymbol{R}_t)^{-1} \big[ \frac{\Delta \boldsymbol{y}_t^c}{\Delta t} - \boldsymbol{H}_t \boldsymbol{m}_t^{fw} \big]$$
$$\simeq \boldsymbol{m}_t^{fw} + \Delta t \boldsymbol{V}_t^{fw} \boldsymbol{H}_t^T \boldsymbol{R}_t^{-1} \big[ \frac{\Delta \boldsymbol{y}_t^c}{\Delta t} - \boldsymbol{H}_t \boldsymbol{m}_t^{fw} \big]$$
$$\hat{\boldsymbol{V}}_t = \boldsymbol{V}_t^{fw} - \Delta t \boldsymbol{V}_t^{fw} \boldsymbol{H}^T (\Delta t \boldsymbol{H}_t \boldsymbol{V}_t^{fw} \boldsymbol{H}_t^T + \boldsymbol{R}_t)^{-1} \boldsymbol{H}_t \boldsymbol{V}_t^{fw}$$
$$\simeq \boldsymbol{V}_t^{fw} - \Delta t \boldsymbol{V}_t^{fw} \boldsymbol{H}^T \boldsymbol{R}_t^{-1} \boldsymbol{H}_t \boldsymbol{V}_t^{fw}$$
$$\boldsymbol{U}_t = (\boldsymbol{I} + \Delta t \boldsymbol{B}_t [\boldsymbol{V}_{t+\Delta t}^{bw}]^{-1})^{-1} (\boldsymbol{I} + \Delta t \boldsymbol{A}_t)$$
$$\simeq \boldsymbol{I} + \Delta t (\boldsymbol{A}_t - \boldsymbol{B}_t [\boldsymbol{V}_{t+\Delta t}^{bw}]^{-1})$$
$$\boldsymbol{v}_t = (\boldsymbol{I} + \Delta t \boldsymbol{B}_t [\boldsymbol{V}_{t+\Delta t}^{bw}]^{-1})^{-1} (\boldsymbol{c}_t + \boldsymbol{B}_t [\boldsymbol{V}_{t+\Delta t}^{bw}]^{-1} \boldsymbol{m}_{t+\Delta t}^{bw})$$
$$\simeq \boldsymbol{c}_t + \boldsymbol{B}_t [\boldsymbol{V}_{t+\Delta t}^{bw}]^{-1} \boldsymbol{m}_{t+\Delta t}^{bw} - \Delta t \boldsymbol{B}_t [\boldsymbol{V}_{t+\Delta t}^{bw}]^{-1} (\boldsymbol{c}_t + \boldsymbol{B}_t [\boldsymbol{V}_{t+\Delta t}^{bw}]^{-1} \boldsymbol{m}_{t+\Delta t}^{bw})$$
$$\boldsymbol{Z}_t \simeq \boldsymbol{B}_t - \Delta t \boldsymbol{B}_t [\boldsymbol{V}_{t+\Delta t}^{bw}]^{-1} \boldsymbol{B}_t.$$

Using the parameterisation from above, the minimisers of the KL divergence are given by

$$\boldsymbol{A}_t^* = \frac{1}{\Delta t} [\langle \Delta \boldsymbol{x}_t \boldsymbol{x}_t^T \rangle - \langle \Delta \boldsymbol{x}_t \rangle \langle \boldsymbol{x}_t \rangle^T][\langle \boldsymbol{x}_t \boldsymbol{x}_t^T \rangle - \langle \boldsymbol{x}_t \rangle \langle \boldsymbol{x}_t \rangle^T]^{-1}$$
$$\simeq \frac{1}{\Delta t}(\boldsymbol{U} - \boldsymbol{I})$$
$$= \boldsymbol{A}_t - \boldsymbol{B}_t [\boldsymbol{V}_{t+\Delta t}^{bw}]^{-1} \tag{31}$$
$$\boldsymbol{c}_t^* = \frac{1}{\Delta t} \langle \boldsymbol{x}_{t+\Delta t} - \boldsymbol{x}_t \rangle - \boldsymbol{A}_t^* \langle \boldsymbol{x}_t \rangle$$
$$\simeq \frac{1}{\Delta t}(\boldsymbol{U} - \boldsymbol{I}) \langle \boldsymbol{x}_t \rangle - \boldsymbol{A}_t^* \langle \boldsymbol{x}_t \rangle + \boldsymbol{v}_t$$
$$\simeq \boldsymbol{c}_t + \boldsymbol{B}_t [\boldsymbol{V}_{t+\Delta t}^{bw}]^{-1} \boldsymbol{m}_{t+\Delta t}^{bw} \tag{32}$$
$$\boldsymbol{B}_t^* = \frac{1}{\Delta t} \left\langle [\Delta \boldsymbol{x}_t - (\boldsymbol{A}_t^* \boldsymbol{x}_t + \boldsymbol{c}_t^*)\Delta t][\Delta \boldsymbol{x}_t - (\boldsymbol{A}_t^* \boldsymbol{x}_t + \boldsymbol{c}_t^*)\Delta t]^T \right\rangle$$
$$\simeq \boldsymbol{B}_t. \tag{33}$$

We remark that when adding the discrete time observations, the form of (31), (32) and (33) does not change, we only have to make sure that the backward filtering for computing $\boldsymbol{m}_t^{bw}$ and $\boldsymbol{V}_t^{bw}$ does include these terms.

## C The inference algorithm

Until convergence do

(1) Update $\{(\boldsymbol{h}^d_{t_i}, \boldsymbol{Q}^d_{t_i})\}_i$ and $\{(\boldsymbol{h}^c_t, \boldsymbol{Q}^c_t)\}_t$ according to

    (1.1) Update $(\boldsymbol{h}^d_{t_i}, \boldsymbol{Q}^d_{t_i})$ by

        (1.1.1) compute the cavity means and variances

$$\boldsymbol{m}^{\backslash t_i}_{t_i} = (\boldsymbol{I} - \boldsymbol{Q}^d_{t_i} \boldsymbol{V}_{t_1})^{-1}(\boldsymbol{m}_{t_i} - \boldsymbol{V}_{t_i} \boldsymbol{h}^d_{t_i}) \quad \text{and} \quad \boldsymbol{V}^{\backslash t_i}_{t_i} = \boldsymbol{V}_{t_i}(\boldsymbol{I} - \boldsymbol{Q}^d_{t_i} \boldsymbol{V}_{t_i})^{-1}$$

        (1.1.2) compute mean $\hat{\boldsymbol{m}}_{t_i}$ and covariance $\hat{\boldsymbol{V}}_{t_i}$ of the tilted distribution $q^i_d(\boldsymbol{x}_{t_i}) \propto p(\boldsymbol{y}^d_{t_i} | \boldsymbol{x}_{t_i}) N(\boldsymbol{x}_{t_i}; \boldsymbol{m}^{\backslash t_i}_{t_i}, \boldsymbol{V}^{\backslash t_i}_{t_i})$ either by exact or numerical methods (see EP related references for details)

        (1.1.3) compute $[\boldsymbol{h}^d_{t_i}]^{new}$ and $[\boldsymbol{Q}^d_{t_i}]^{new}$ from

$$[\boldsymbol{h}^d_{t_i}]^{new} = [\hat{\boldsymbol{V}}_{t_i}]^{-1}\hat{\boldsymbol{m}}_{t_i} - [\boldsymbol{V}^{\backslash t_i}_{t_i}]^{-1}\boldsymbol{m}^{\backslash t_i}_{t_i}$$
$$[\boldsymbol{Q}^d_{t_i}]^{new} = [\hat{\boldsymbol{V}}_{t_i}]^{-1} - [\boldsymbol{V}^{\backslash t_i}_{t_i}]^{-1}$$

    (1.2) Update $\boldsymbol{h}^c_t$ and $\boldsymbol{Q}^c_t$ by

$$[\boldsymbol{h}^c_t]^{new} = -\partial_{\boldsymbol{m}_t} \langle V(t, \boldsymbol{x}_t) \rangle_{\mathcal{N}(\boldsymbol{m}_t, \boldsymbol{V}_t)} + 2\partial_{\boldsymbol{V}_t} \langle V(t, \boldsymbol{x}_t) \rangle_{\mathcal{N}(\boldsymbol{m}_t, \boldsymbol{V}_t)} \boldsymbol{m}_t$$
$$[\boldsymbol{Q}^c_t]^{new} = \partial_{\boldsymbol{V}_t} \langle V(t, \boldsymbol{x}_t) \rangle_{\mathcal{N}(\boldsymbol{m}_t, \boldsymbol{V}_t)}$$

(2) Update $\boldsymbol{m}_t$ and $\boldsymbol{V}_t$ by

    (2.1) solve forward starting at $(\boldsymbol{m}_0, \boldsymbol{V}_0)$

$$\frac{d}{dt}\boldsymbol{V}^{fw}_t = \boldsymbol{A}_t\boldsymbol{V}^{fw}_t + \boldsymbol{V}^{fw}_t\boldsymbol{A}^T_t + \boldsymbol{B}_t - \boldsymbol{V}^{fw}_t\boldsymbol{Q}^c_t\boldsymbol{V}^{fw}_t, \qquad \boldsymbol{m}^{fw}_{t_i+} = (\boldsymbol{I} + \boldsymbol{V}^{fw}_{t_i}\boldsymbol{Q}^d_{t_i})^{-1}(\boldsymbol{m}^{fw}_{t_i} + \boldsymbol{V}^{fw}_{t_i}\boldsymbol{h}^c_{t_i}),$$
$$\frac{d}{dt}\boldsymbol{m}^{fw}_t = \boldsymbol{A}_t\boldsymbol{m}^{fw}_t + \boldsymbol{c}_t + \boldsymbol{V}^{fw}_t[\boldsymbol{h}^c_t - \boldsymbol{Q}^c_t\boldsymbol{m}^{fw}_t], \qquad \boldsymbol{V}^{fw}_{t_i+} = (\boldsymbol{I} + \boldsymbol{V}^{fw}_{t_i}\boldsymbol{Q}^d_{t_i})^{-1}\boldsymbol{V}^{fw}_{t_i}$$

    (2.2) solve backwards starting, say, at $(\boldsymbol{0}, 100\boldsymbol{V}^{fw}_1)$

$$\frac{d}{dt}\boldsymbol{V}^{bw}_t = \boldsymbol{A}_t\boldsymbol{V}^{bw}_t + \boldsymbol{V}^{bw}_t\boldsymbol{A}^T_t - \boldsymbol{B}_t + \boldsymbol{V}^{bw}_t\boldsymbol{Q}^c_t\boldsymbol{V}^{bw}_t, \qquad \boldsymbol{m}^{bw}_{t_i-} = (\boldsymbol{I} + \boldsymbol{V}^{bw}_{t_i}\boldsymbol{Q}^d_{t_i})^{-1}(\boldsymbol{m}^{bw}_{t_i} + \boldsymbol{V}^{bw}_{t_i}\boldsymbol{h}^d_{t_i}),$$
$$\frac{d}{dt}\boldsymbol{m}^{bw}_t = \boldsymbol{A}_t\boldsymbol{m}^{bw}_t + \boldsymbol{c}_t - \boldsymbol{V}^{bw}_t[\boldsymbol{h}^c_t - \boldsymbol{Q}^c_t\boldsymbol{m}^{bw}_t], \qquad \boldsymbol{V}^{bw}_{t_i-} = (\boldsymbol{I} + \boldsymbol{V}^{bw}_{t_i}\boldsymbol{Q}^d_{t_i})^{-1}\boldsymbol{V}^{bw}_{t_i}$$

    (2.3) compute $\boldsymbol{m}_t$ and $\boldsymbol{V}_t$ from

$$[\boldsymbol{V}_t]^{-1} = [\boldsymbol{V}^{fw}_t]^{-1} + [\boldsymbol{V}^{bw}_t]^{-1} \quad \text{and} \quad \boldsymbol{m}_t = \boldsymbol{V}_t[[\boldsymbol{V}^{fw}_t]^{-1}\boldsymbol{m}^{fw}_t + [\boldsymbol{V}^{bw}_t]^{-1}\boldsymbol{m}^{bw}_t]$$

The above equations are given for illustrative purposes only, one should always avoid inversion, try to stabilise computations by reorganising them and, when necessary, performing matrix inversions through matrix factorisations.

The sequential forward-backward scheduling follows by iteratively solving the forward and backward equations with $\boldsymbol{Q}^c_t$ and $\boldsymbol{h}^c_t$ computed according to (1.2) and the update steps (1.1) for $\boldsymbol{Q}^d_t$ and $\boldsymbol{h}^d_t$ performed at the jump times $t_i$.