[Reviews · NeurIPS 2013]

Submitted by Assigned_Reviewer_6

-- Update
Just a few other thoughts. Simo Saarka has more recent work on continuous-discrete time systems (as you referenced) that might be interesting to contrast against - there, using Gaussian cubature that provides a nice alternative deterministic approximation method. This might be useful for additional discussion and future work. In addition I also now wondered if it is possible to derive an algorithm directly using the variational Gaussian approach. This would be more appealing from the point of having a well defined objective function with which to optimise, potentially fewer numerical issues and interpretation directly in terms of the marginal likelihood. We could afterwards add low-order marginal corrections using cumulant perturbations (like those of Opper for EP) - the only place I think shows this is Barber and van de Laar (http://arxiv.org/pdf/1105.5455.pdf). I do look forward to reading the final version of the paper.

-- Original
The paper presents an algorithm for approximate Bayesian inference in models
with continuous and discrete time observations. The model can be
cast in the framework of latent Gaussian models and a parallel
expectation propagation algorithm can be used to derive a principled approach for
inference and learning dealing with both continuous and discrete time. This EP inference algorithm is
embedded within an EM algorithm to both learn parameters of the model as
well as marginal distributions. The algorithm is shown to be effective in
the number of experimental settings.

Overall I enjoyed the paper and thoughts that it extends the
applicability of approximately message passing to a wider class of models.

In particular I thought it was interesting that EP updates for a
continuous time limit collapsed to the variational Gaussian updates. This is
related to the latent Gaussian structure but I wondered if there is a
deeper reason underlying this connection.

The algorithm seems robust due to be implied fractional updating but
I wondered if you could comment on any experienced
difficulties in implementation, such as issues of slow convergence of parameter learning,
numerical stability, etc.

The algorithm is still cubic due to the inverses is in the inference
as well as the M-step updating - could comment on approaches to scaling up such algorithms.

In the experimental section it would be nice to see plots giving
insight into the convergence of the algorithm. Can we also demonstrate the advantages
obtained by of having an estimate of the marginal likelihood.
For example, it could be possible in figure 3C to plot
each of the individual points with a size proportional the marginal likelihood value.

Summary: Overall the paper is well written and extend the applicability of
approximate Bayesian inference methods to the class of continuous and
discrete time settings, which many will find interesting.

Submitted by Assigned_Reviewer_10

This paper applies approximate inference to nonlinear diffusion equations by taking the continuous-time limit of the expectation-propagation technique. The result is a tracking algorithm which is, naturally, much faster than sampling methods, and on the experiments shown is rather accurate. I'm curious if a linearization of the loss function + the application of Kalman-Bucy (ie, extended KB), possibly applied iteratively, would lead to a more/less effective algorithm. It would also be interesting to see more substantial experiments, for instance with high-frequency financial data where this framework is often use and existing benchmarks are available.

Quality: the paper is technically solid.
Clarity: the paper is clearly written and well organized.
Originality: relatively high.
Significance: the speedup over MC achieved here is potentially important.
Summary: This paper applies an approximate inference method, namely, expectation-propagation, to nonlinear diffusion processes and obtains a significant speedup over sampling methods. The paper is clear and straightforward to follow.

Submitted by Assigned_Reviewer_11

This is a theoretically strong and interesting paper that proposes a novel EP-type inference approach for continuous-time stochastic dynamical systems. The paper is well structured and the theory clearly presented. However, the authors should have better motivated the approach with a stronger experimental section. The only comparison with other approaches is in the first example in Section 3.1, where the authors use a MCMC approach as a benchmark. A part from not completely agreeing in using MCMC approaches as a benchmark, as the proposed method performs as well as the MCMC approach, what is the advantage in using it? The authors should discuss
this point in detail. In the third example, the authors seem to suggest that the results are similar to those in Zammit Mangion et al. They then justify the use of their approach from a computational viewpoint. The authors should give some quantification of the advantage -- it is not useful for the community to introduce a new approach without giving an idea of its characteristics with respect to existing ones. I would really appreciate some quantitative answers on this point in the rebuttal period.
Summary: Very interesting paper but the experimental section is not completely satisfactory.
Author Feedback

Author rebuttal: We thank the reviewers for their appreciative comments and constructive criticism, please find the corresponding replies below.

Assigned reviewer 10:
The suggested algorithm is interesting, as would be interesting comparisons with a local Laplace approximation (which is clearly related); implementation of these approaches in the continuous time setting is non-trivial and best left for further future comparisons. From experience in multivariate and discrete time models, EP and variational tend to perform better than either extended Kalman filtering or Laplace approximations (Ypma&Heskes, 2005; Zoeter&Ypma&Heskes, 2006), which leads to suspect these comparisons would also hold in continuous time. We appreciate the suggestion of financial data as a good arena for testing the method, but, as we are not domain experts, we do not feel we can properly address that at the moment. However, we will definitely consider such models and fields of application in the future as well as various other inference algorithms and correction schemes.

Assigned reviewer 11:
We agree with the reviewer that MCMC is also an approximation, however it is frequently used in ML as a benchmark, so we followed that approach. The advantages of the proposed method are both in computational speed and in the retained continuous time nature of inference (as we point out, MCMC needed time discretisation and sampling at small time-lag can be computationally demanding (Golightly&Wilkinson, 2008; Kou et al.,2012). Similar considerations also hold for the comparison with Zammit-Mangion et al; we emphasised the similarity of some results because of their biological interpretability (clustering of parameters). Some of the advantaged of continuous time approach are: (i) numerical stability when time-lag tends to zero (ii) a natural way of dealing with non-equidistant observations or observations that are too close to each other in time (iii) a natural interpretability of parameters. We will expand the discussion of the computational advantages in the final version; however, we emphasise that the contributions of the paper are a novel methodology for continuous time systems.

Assigned reviewer 6:
We also wonder whether large scale averaging effects underlie the collapse of EP to variational, but we have no further proof of this intuition. Formally, the collapse happens due to the time-lags limiting procedure (or the cavity distribution being equal in limit to the marginal). There are indeed issues with slow EM convergence on the real-world data set, these can be remedied using an expected conjugate gradient framework (Salakhutdinov et al. ICML-2013), however, we opted for EM because of the ease in exposition, suitability to the framework of the VB-EM. The convergence of the method is typical to the fixed point iteration or message passing methods, with an sensible initialisation around 10-20 iterations (forward-backwards) were sufficient. Clearly, as with any EP type algorithms, multimodal and non-log-concave losses and likelihoods can cause issues and other avenues such as double loop (Yuille et al., 2002 or Heskes&Zoeter, 2002) and direct minimisation (Welling&Tech2001, Archambeau 2007) have to be explored. The cubic nature of the algorithm is inherent in the Riccati differential equations for the covariance parameters (even with sparse A_t and a diagonal B_t) and we are not aware of any methods in the literature that could do a decent job in continuos time models as for example Cseke et al. (2013) (http://arxiv.org/abs/1305.4152) in discrete time models. Nonetheless, we are actively exploring alternatives, one possibility is using path-wise decompositions and LBP type approaches over paths in distributed systems. Thank you the interesting suggestion w.r.t. panel 3c, we will try to find a way to scale the points in the final version so that the panel remains visually pleasing.